# SKIM: Any-bit Quantization Pushing the Limits of Post-Training Quantization

**Runsheng Bai** [1]   **Bo Liu** [2]   **Qiang Liu** [2]

## Abstract

Large Language Models (LLMs) exhibit impressive performance across various tasks, but deploying them for inference poses challenges. Their high resource demands often necessitate complex, costly multi-GPU pipelines, or the use of smaller, less capable models. While quantization offers a promising solution utilizing lower precision for model storage, existing methods frequently experience significant performance drops at lower precision levels. Additionally, they typically provide only a limited set of solutions at specific bit levels, many of which are extensively manually tuned. To address these challenges, we propose a new method called **SKIM**: Scaled K-means clustering wIth Mixed precision. Our approach introduces two novel techniques: 1. A *greedy algorithm* to solve approximately optimal bit allocation across weight channels, and 2. A *trainable scaling vector* for non-differentiable K-means clustering. These techniques substantially improve the model performance and can be adapted to any given bit. Notably, in terms of perplexity, our method narrows the gap between quantized LLaMA models and their full precision counterparts by around **14%** on average.

## 1. Introduction

Large Language Models (LLMs) including GPT (Radford et al., 2019) and LLaMA (Touvron et al., 2023a), have achieved remarkable performance across a diverse range of tasks. These models not only excel in language processing (Brown et al., 2020; Dubey et al., 2024; Chowdhery et al., 2023; Zhang et al., 2022) but also adapt effectively to multimodal applications (Wang et al., 2024; Driess et al., 2023), marking a crucial step toward artificial general in-

---
[1]EECS Department, Massachusetts Institute of Technology [2]Department of Computer Science, University of Texas at Austin. Correspondence to: Qiang Liu <lqiang@cs.utexas.edu>, Runsheng Bai <runsheng@mit.edu>.

*Proceedings of the $42^{nd}$ International Conference on Machine Learning*, Vancouver, Canada. PMLR 267, 2025. Copyright 2025 by the author(s).

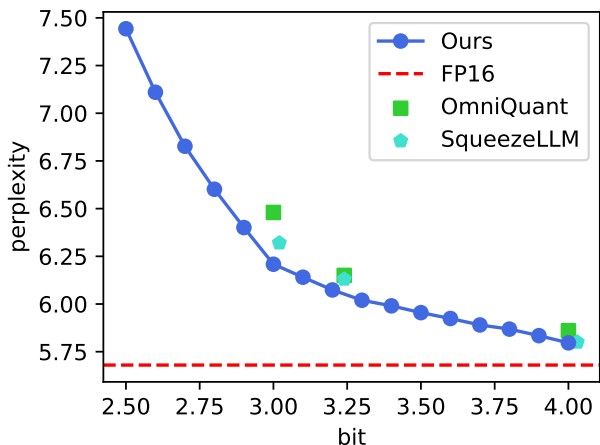

*Figure 1.* Our SKIM method adaptively quantizes the model to any specified bit and achieves superior performance. The perplexity reported is of LLaMA-7B on the WikiText2 dataset.

telligence (Bubeck et al., 2023). However, the computational and memory demands of LLMs pose significant challenges. For instance, when loading parameters in FP16, GPT requires 350GB of memory, while LLaMA-65B needs at least 130GB, both of which far exceed the capabilities of A100-80G GPUs. Even when conducting inference with the smallest LLaMA model (7 billion parameters), an Out-of-Memory exception can occur on a widely used 24GB GPU. These challenges significantly complicate the storage and practical deployment of such models.

One promising technique to address these issues is quantization, which involves transforming high-precision data into lower-precision formats, such as converting FP16 parameters to INT4. This method directly reduces memory requirements for deploying and loading LLMs, while improving inference latency by alleviating the memory bandwidth bottleneck, also known as the memory wall (Gholami et al., 2024). In addition, quantization has shown promising performance benefits. For example, previous studies have shown that both LLM weights and activations can be stored in 8 bits (Xiao et al., 2023), or only LLM weights can be stored in 4 bits (Kim et al., 2023), with little performance degradation. This encourages researchers to explore lower-precision solutions while maintaining reasonable performance levels.

However, standard quantization techniques in recent meth-

ods often suffer a significant drop in performance when using low bit widths. To mitigate this decline, these methods often introduce additional techniques that incur extra memory costs. For example, SqueezeLLM (Kim et al., 2023) retains certain sensitive elements and outliers with full precision using a sparse tensor, while AWQ (Lin et al., 2024) divides the quantization group into smaller ones, requiring the storage of more quantization factors. Additionally, the extra memory needed to achieve a reasonable trade-off between memory usage and performance often requires manual tuning and selection, making the process cumbersome.

**Contribution** In this paper, we address the above issues with our proposed method, Scaled K-means clustering wIth Mixed Precision (SKIM), which optimizes the bit allocation using a greedy algorithm and regularizes the column-wise difference with a scaling vector. Our method can easily adapt to any specified bit, including even non-integer values, and achieve better performance. Figure 1 illustrates how our method breaks the fixed bit grid and delivers better results. Our main contributions can be summarized as follows: (1) We conduct a mathematical analysis of two optimization targets: layer-wise and sensitivity-based quantization, identifying a unified framework that highlights their core differences and allows us to evaluate their effectiveness. (2) We observe a significant disparity in data distribution across channels and propose a greedy algorithm for approximately optimal bit allocation in response to this disparity. Our mixed-precision method supports any bit level and demonstrates significant performance gains. (3) For the non-differentiable K-means clustering operator, we incorporate a trainable scaling vector based on our novel iterative optimization strategy. This vector effectively regularizes the data across columns and serves as a valuable complement to the mixed precision method.

## 2. Related Work

**Quantization of LLMs** Quantization can be viewed from different perspectives. It can be categorized into two types based on whether the entire model is trained with quantization in mind: *Quantization-Aware Training (QAT)* (Jacob et al., 2018; Xi et al., 2023) and *Post-Training Quantization (PTQ)* (Cai et al., 2020; Shomron et al., 2021). While QAT methods typically perform better, their high resource requirements for retraining make them less practical for LLMs, making PTQ the preferred choice for language models and the focus of our work.Quantization methods can also be classified into *Weight-Activation Quantization* (Yao et al., 2022; Xiao et al., 2023) and *Weight-Only Quantization* (Frantar et al., 2022; Chee et al., 2023; Lin et al., 2024), depending on whether both weights and activations are quantized. This paper focuses on the Weight-Only method, which directly addresses demands for deploying the model.

**Non-uniform quantization** Non-uniform quantization (Jeon et al., 2022; Liu et al., 2022b) uses varying quantization intervals to better align with data distribution, leading to improved performance. Among various techniques, such as space transformation (Yvinec et al., 2023) and trainable quantization factors (Jeon et al., 2022), K-means clustering (Krishna & Murty, 1999; Kanungo et al., 2002) is widely adopted (Xu et al., 2018; Zadeh et al., 2020; Kim et al., 2023). It generates cluster labels and centroids, enabling us to store the labels directly as low-bit data and the centroids as a codebook for recovery.

**Quantization Techniques** Outliers have been a significant obstacle for LLM quantization in achieving lossless solution (Dettmers et al., 2022; Xiao et al., 2023). Previous works have proposed various methods to address this issue, some of which share similarities (Dettmers et al., 2023; Xiao et al., 2023) with our mixed precision technique and scaling vector. However, our approach differs notably. Existing mixed precision techniques typically combine a specific bit level with the original precision, like INT3 and FP16, using element-wise mixing to preserve crucial elements. In contrast, our method adaptively blends all available bit levels and uses a channel-wise mixture for better resource allocation. Regarding scaling factors, most existing methods apply them in uniform quantization, considering only differential operators. Our work shifts the focus to a non-uniform context, optimizing scaling on non-differential grouping operators through a novel strategy.

## 3. Review The Quantization Objectives

Previous works have proposed different quantization objectives, significantly broadening the scope of this field. However, these objectives are often considered in isolation, making the evaluation and selection of objectives unnecessarily complicated, especially for our multi-process SKIM method. In this section, we conduct a comparative analysis on two widely adopted approaches: the layer-wise and sensitivity-based objectives. This analysis highlights their similarities and key distinctions, providing foundations for informed selection, which will be discussed in Section 4.1.

### 3.1. Notations

We define a general linear layer using following notations:

- $W \in \mathbb{R}^{n \times m}$ and $X \in \mathbb{R}^{m \times k}$ denote the weight and input matrix, respectively. And the corresponding output matrix is $Y = WX \in \mathbb{R}^{n \times k}$.

- The quantized weight, denoted as $W^q \in \mathbb{R}^{n \times m}$, is the full-precision matrix reconstructed from its low-bit representation. It satisfies the constraint that its values are restricted to $2^{bit}$ discrete centroids for each group.

- The $i$-th row of $W$ is represented as $w_i \in \mathbb{R}^{1 \times m}$. The same definition applies to $w_i^q$, $x_i$ and $y_i$.

- The gradient and Hessian matrix are computed with respect to the final loss $L$. Taking $w_i$ as example, $g_{w_i} = \nabla_{w_i} L \in \mathbb{R}^{1 \times m}$ represents the gradient, and $H_{w_i} = \nabla_{w_i}^2 L \in \mathbb{R}^{m \times m}$ represents the Hessian matrix.

## 3.2. Layer-wise Quantization

The Layer-wise Quantization Framework has been widely adopted (Frantar et al., 2022; Hubara et al., 2021) to make the task more targeted. This framework aims to quantize each layer individually and addresses corresponding reconstruction problems. Concretely, let $W$ be the full-precision weight matrix, and $X$ the input data. The goal is to find the quantized weight $W^q$ that minimizes the layer-wise squared error between the outputs of the original and quantized weights, which can be formally expressed as:

$$\arg \min_{W^q} \| WX - W^q X \|^2. \tag{1}$$

## 3.3. Sensitivity-based Quantization

Instead of minimizing the layer-wise squared error, SqueezeLLM (Kim et al., 2023) proposes minimizing the overall perturbation with respect to the final loss. They use the second-order Taylor expansion to analyze how changes in a specific layer weight $W$ influence the final loss, and further assume the first-order term is approximately zero since the model to be quantized should have already converged. For simplicity in understanding, here we use $w_i$, the $i$-th row of $W$, to explain the target. With its Hessian matrix $H_{w_i}$, the objective can be written as:

$$\arg \min_{w_i^q} (w_i - w_i^q) H_{w_i} (w_i - w_i^q)^\top. \tag{2}$$

Furthermore, two additional approximations are incorporated: 1. Take Fisher information matrix computed on a calibration dataset $D$ as a Hessian approximation to avoid heavy computation. It can be formally expressed as $H \approx F = \mathbb{E}(g^\top g) \approx \frac{1}{|D|} \sum_{d \in D} (g^{(d)})^\top g^{(d)}$, where $F$ is the Fisher information matrix and $g^{(d)}$ represents the gradient computed on sample $d$; 2. The Fisher information matrix is further approximated as a diagonal matrix by assuming that cross-weight interactions are negligible, reducing the complexity from quadratic to linear. Consequently, with $\mathrm{diag}(\cdot)$ representing the diagonal function, the final objective can be written as:

$$\arg \min_{w_i^q} (w_i - w_i^q) \mathrm{diag}(\mathcal{F}_{w_i}) (w_i - w_i^q)^\top \tag{3}$$

$$= \arg \min_{w^q} \sum_{j=1}^{m} (g_{w_i})_j^2 \cdot (w_{ij} - w_{ij}^q)^2. \tag{4}$$

## 3.4. Reformulation

Although layer-wise and sensitivity-based objectives have different assumptions and initial forms, the following reformulation aims to organize them into unified structures, either *full* or *diag*, which will be discussed later.

**Layer-wise quantization**   Since the L2-norm of a matrix can be expressed as the sum of that of its rows, Equation (1) can be reformulated in terms of the $i$-th row of $W$:

$$\arg \min_{w_i^q} \| (w_i - w_i^q) X \|^2 \tag{5}$$

$$= \arg \min_{w_i^q} (w_i - w_i^q) X X^\top (w_i - w_i^q)^\top. \tag{6}$$

Here $XX^\top$ happens to be the Hessian approximation derived from the Fisher information matrix with respect to the layer-wise target, sharing the same structure as Equation (2). In the following discussion, we refer to this objective as *L-full*. Similarly, we can also take the diagonal approximation here, by assuming that $X$ has an expectation close to zero. We use *L-diag* to refer to this approximated objective:

$$\arg \min_{w_i^q} (w_i - w_i^q) \mathrm{diag}(XX^\top) (w_i - w_i^q)^\top \tag{7}$$

$$= \arg \min_{w^q} \sum_{j=1}^{m} \| x_j \|^2 \cdot (w_{ij} - w_{ij}^q)^2. \tag{8}$$

**Sensitivity-based quantization**   For the $i$-th row of $W$ and $Y$, the relationship $y_i = w_i X$ holds. Consequently, the gradient relationship can be expressed as $g_{w_i} = g_{y_i} X^\top$. Using this result, Equation (2) with Fisher approximation can be rewritten as:

$$\arg \min_{w_i^q} (w_i - w_i^q) g_{w_i}^\top g_{w_i} (w_i - w_i^q)^\top \tag{9}$$

$$= \arg \min_{w^q} \left( (w_i - w_i^q) X g_{y_i}^\top \right)^2. \tag{10}$$

This Equation shares the same form as Equation (5) and the only difference between them is whether include $g_{y_i}$ is included. It be referred to as *S-full* in the following text. Similarly, Equation (4) can also be expressed in a form that includes the output gradient and input:

$$\arg \min_{w^q} \sum_{j=1}^{m} \left( x_j \cdot g_{y_i}^\top \right)^2 \cdot (w_{ij} - w_{ij}^q)^2. \tag{11}$$

This form is structurally identical to Equation (8) and will be denoted as *S-diag*.

To conclude, the above transformation highlights the similarities and core differences between layer-wise and sensitivity-based quantization, allowing us to intuitively evaluate their effectiveness and make selections, as detailed in Section 4.1 under the context of our SKIM method.

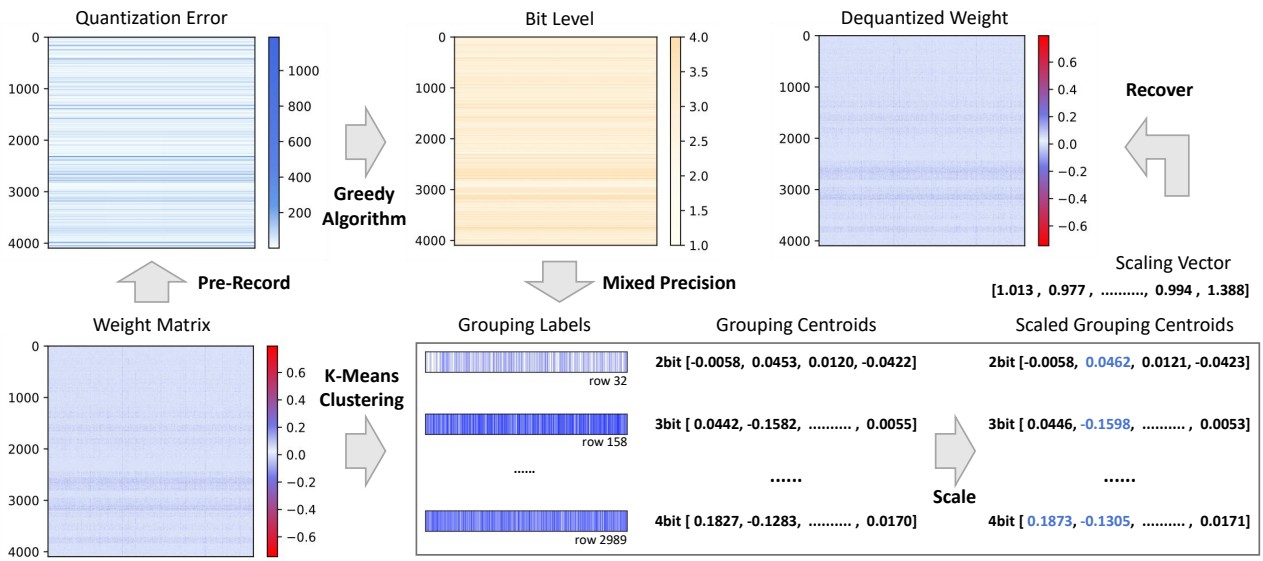

*Figure 2.* Overall procedure of our proposed SKIM algorithm. The method consists of three main part: greedy algorithm for bit allocation, weighted K-Means Clustering based on allocation, and the trainable scaling vector. More details are available in Section 4.

## 4. Methodology

The entire process of our SKIM algorithm is detailed in Figure 2. It begins with a greedy algorithm that allocates different bits to each channel based on pre-recorded quantization errors. If channel $i$ is allocated $b_i$ bits, $2^{b_i}$ centroids will be generated for it. Next, weighted K-means clustering is applied to compute the centroids and labels for each channel. Finally, we incorporate the scaling vector and train it through an iterative optimization strategy. The dequantized weights can be recovered from the final labels, centroids, and scaling vector. Full algorithm of our SKIM method is illustrated in Appendix A. In the following subsections, we will detail our mixed precision and scaling vector techniques, as well as the principles behind our objective selection.

### 4.1. Objective Selection

As discussed, our SKIM framework consists of three main steps: the greedy algorithm, weighted K-Means clustering, and the scaling vector. This multi-step process raises some questions: Should the same objective be used consistently across all steps, or is cross-objective optimization feasible? Moreover, under different computational scenarios, which objective is most effective, and how should it be selected?

These questions can be addressed through our previous reformulations, from which we can conclude that, although the layer-wise and sensitivity-based objectives differ, they can ultimately be transformed into either the *full* or *diag* forms. In these forms, both objectives share a similar structure, differing only in the introduction of $g_{y_i}$ to guide the subsequent model architecture. This alignment ensures their

synchronization towards the final goal and facilitates cross-objective optimization in our work. Related experimental results are provided in Section 5.4.2.

Therefore, we can intuitively assess the effectiveness of each objective in the following order, from best to worst: *S-full*, *L-full*, *S-diag*, and *L-diag*. This ranking reflects the fact that the *S* form incorporates gradient information as a guide, while the *diag* form takes an aggressive diagonal approximation. This analysis aligns with our experimental results, which will be detailed in Section 5.4.1.

However, due to the interdependence among the elements of the gradient and the input, we cannot compute the four objectives using directly recorded expectations of $g$ or $X$. This limitation prevents us from adopting the most effective *S-full* objective, as recording the corresponding $\mathbb{E}(g^\top g)$ for each row requires quadratic memory complexity which is impractical for LLMs.

Consequently, in any scenario requiring a complete error calculation, such as the proposed mixed precision and scaling vector methods, we adopt the *L-full* form. In contrast, for scenarios involving element-wise sums, such as the weighted K-means clustering, we adopt the *S-diag* form. This will be the main principle for our objective selection.

### 4.2. Mixed Precision

As previously mentioned, the quantization error of a weight matrix $W$ is the sum of that of its individual rows. However, each row, representing each channel, often exhibits distinct data distributions and quantization errors, as illustrated in

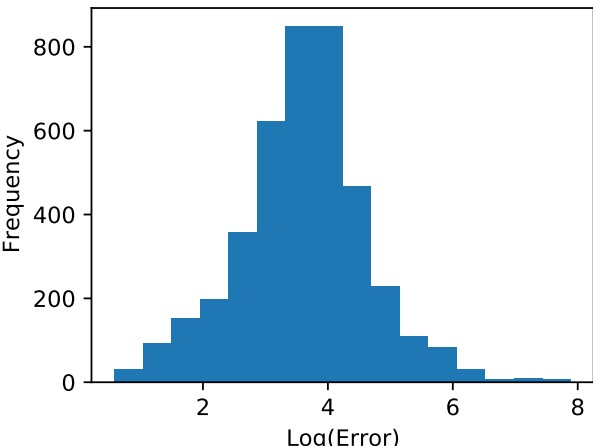

*Figure 3.* Histogram of the channel-wise quantization error for the $self\_attn.q\_proj$ in the second layer of Llama-7B. Errors vary significantly and exhibits a long-tail distribution on the larger side.

Figure 3. Additionally, Figure 4 demonstrates that the quantization error after a one-bit increase is also unpredictable, even when the error at the current bit is known. These observations indicate that applying the same bit-width for quantizing all rows results in a disproportionate allocation of resources. Motivated by these insights, we propose the adaptive channel-wise **Mixed Precision** technique, which aims to solve:

$$\underset{b_1,b_2,...,b_n}{\arg\min} \sum_{i=1}^{m} Err(w_i, b_i), \ s.t. \ \frac{1}{n}\sum_{i=1}^{m} b_i \leq \hat{b}, \quad (12)$$

where $b_i$ represents the bit allocated for channel $i$, $\hat{b}$ is the total bit constraint, and $Err(\cdot)$ function refers to the *L-full* form of error between original $w_i$ and $b_i$ bit quantized weight, as analyzed earlier. This formulation identifies the mixed precision issue as a bit-constrained sum minimization problem, which can be viewed as a variation of the knapsack problem (Martello & Toth, 1987; Cacchiani et al., 2022) and precisely solved using a dynamic programming algorithm. However, the quadratic computational complexity of this algorithm renders it infeasible for scalability in

---

**Algorithm 1** Algorithm for Bit Allocation

---

**input** Error Matrix $E$, Minimum Available Bit $b_{\min}$, Maximum Available Bit $b_{\max}$, Row Number $n$, Bit $bit$
**output** Bit Allocation $b$
    Initialize $b = \{b_{\min}\}_n \in \mathcal{R}^n$
    **repeat**
        $i = \arg\max_{0 \leq i < n}\{E_{i,b_i+1} - E_{i,b_i}| \ b_i < b_{\max}\}$
        $b_i = b_i + 1$
    **until** $\sum_{i=1}^{n} b_i \geq n \cdot bit$
    **return** $b$

---

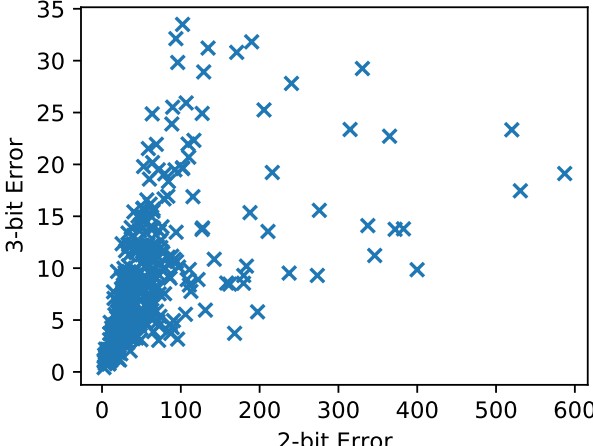

*Figure 4.* Error variation of the $self\_attn.q\_proj$ in the first layer of Llama-7B. We randomly sampled 10% of the total rows for clearer visualization, with each point representing one. The horizontal axis indicates the quantization error when using 2 bits, while the vertical axis shows the error after increasing the bit level from 2 to 3. It is important to note that after the increase, same previous quantization error does not imply a similar post-increase error, and larger error does not lead to a larger result as well.

LLMs. Therefore, we opt for a faster greedy algorithm to approximate the optimal solution. The greedy algorithm can yield a solution sufficiently tight to the optimal one while only operates with a time complexity of $O(n \log n)$". Similar greedy algorithms have been widely adopted in various topics related to language models (Chen et al., 2021; Liu et al., 2022a), demonstrating its effectiveness. Algorithm 1 describes the specific steps in detail.

### 4.3. Scaling Vector

Inspired by previous works (Xiao et al., 2023), we recognized that a scaling vector could effectively regularize differences across channels, complementing our mixed-precision method. This operation can be formally expressed as:

$$W^q = Quant(W * \alpha^{-1}) * \alpha,$$

where $W$ is the original weight, $*$ represents the element-wise multiplication with broadcasting, and $\alpha^{-1}$ indicates the element-wise reciprocal of the scaling vector.

However, unlike previous works, a significant challenge in our scenario lies in determining the values of the scaling vector $\alpha$. Outliers play a less dominant role in weight distributions compared to activations, and under channel-wise quantization, the most intuitive approaches based on similarity measures cannot be directly applied. Empirical values suggested by SmoothQuant (Xiao et al., 2023) and similar choices fail in our setting, leading to counterintuitive increases in perplexity. This highlights the need for a robust,

automated approach to derive scaling values, rather than relying on fixed heuristics.

An intuitive solution is to make the scaling vector learnable and optimize it toward a reasonable target, similar to OmniQuant (Shao et al., 2023) which employs gradient descent on block-wise error to train equivalent transformations. However, the presence of a non-differentiable grouping operator in K-means clustering precludes direct gradient descent on $\alpha$. To tackle this challenge, we propose a novel strategy called iterative optimization, which optimizes grouping and scaling separately. By keeping the grouping results fixed during training, this approach enables effective gradient computation for optimizing $\alpha$. Algorithm 2 provides a detailed explanation of the strategy.

---

**Algorithm 2** Iterative Optimization

---

**input** Weight $W$, Iteration $I = 1$
**output** Labels $L$, Centroids $c$, Scaling Vector $\alpha$
  $n, m = W.shape$
  Initialize $\boldsymbol{\alpha} = \mathbf{1}_{1 \times m} \in \mathcal{R}^{1 \times m}$
  **for** $i = 1$ **to** $I$ **do**
    $\tilde{W} = W * \alpha^{-1}$        *// element-wise multiplication*
    $L = \mathbf{0}_{n \times m} \in \mathcal{R}^{n \times m}$
    **for** $i = 1$ **to** $n$ **do**
      *// optimize on grouping*
      $L_{i,:} \leftarrow Kmeans(\tilde{W}_{i,:}).labels$
    **end for**
    **repeat**
      $c = calc\_centroids(W * \boldsymbol{\alpha}^{-1}, L)$
      $W^q = replace(c, L) * \boldsymbol{\alpha}$
      $loss = Err(W, W^q)$
      $loss.backward()$        *// optimize on scaling*
    **until** Converged
  **end for**
  **return** $L, c, \alpha$

---

Note that Algorithm 2 is a simplified version of our SKIM method, clarifying the functionality of iterative optimization; the full SKIM method is illustrated in Appendix A. The loss calculation adopts the *L-full* form of error, as discussed in our previous analysis, while the optimization of grouping corresponds to the previously mentioned K-means Clustering. When mixed precision is enabled, the $Kmeans(\cdot)$ function uses the allocation results from Algorithm 1, quantizing each channel with different precisions.

## 5. Experiments

### 5.1. Setups

**Quantization Details**  We evaluate our method within the context of post-training and weight-only quantization. The default setting includes INT4 and INT3, as well as INT3 and INT2 with extra memory usage. Note that we have set the maximum available bit to 4 in order to maintain high memory efficiency. Consequently mixed precision is disabled under the INT4 setting. And to optimize the scaling vector, we utilize the Adam (Kingma, 2014) optimizer with a learning rate of 0.01, a decrease rate of 0.5 every 40 steps and a maximum number of iterations of 120.

**Baselines**  We primarily compare our method against three baselines: SqueezeLLM (Kim et al., 2023), Omni-Quant (Shao et al., 2023) and QuIP# (Tseng et al., 2024) . SqueezeLLM provides state-of-the-art performance under INT4 and INT3 settings, while OmniQuant offers greater flexibility and performs better in the INT2 setting. QuIP# achieves comparable SOTA performance as well; however, given the increased resource demands associated with its finetuned version, we opt to use QuIP# without finetuning as one of our baselines. And since most existing works offer limited available bit levels, we ensure fairness by aligning our chosen bit levels with those that are more widely adopted and only comparing under similar conditions. Comparison with other baselines, including DecoupleQ (Guo et al., 2024) and ABQ-LLM (Zeng et al., 2024), can be found in Appendix B.2.

**Models and Datasets**  We evaluate our method across various models, including LLaMA (Touvron et al., 2023a), LLaMA2 (Touvron et al., 2023b), and OPT (Zhang et al., 2022), to assess its generalizability. Results for the LLaMA models are emphasized in the main text due to their widespread adoption, while comprehensive results for other models can be found in Appendix B.1. Regarding datasets, we primarily utilize WikiText2 (Merity et al., 2016) and C4 (Raffel et al., 2020) for evaluation, along with 100/128 samples from the C4 dataset for calibration.

**Evaluation**  We use the perplexity of language generation experiments as a primary metric, reporting results on the WikiText2 (Merity et al., 2016) and C4 (Raffel et al., 2020) datasets. Since our calibration dataset is derived from C4, perplexity on WikiText2 represents a zero-shot scenario, while perplexity on C4 corresponds to a few-shot scenario. We omit specific average bits under integer settings due to negligible differences between methods, opting instead for reporting unified bit levels for clarity. Additionally, we evaluate accuracy on all PiQA(Bisk et al., 2020), ARC-Challenge/Easy(Clark et al., 2018) and MMLU (Hendrycks et al., 2020) benchmarks under both INT3/INT4 settings to assess the problem-solving capability of our quantized model. Since none of the calibration data directly relates to these benchmarks, the results may represent a complete zero-shot scenario and provide a high-level representation of our method's knowledge maintenance capability.

*Table 1.* **Quantization Result of LLaMA models.** We report perplexity of quantized LLaMA-7B and LLaMA2-7B in this table. Note that since SqueezeLLM did not provide an kernel implementation for 2-bit setting, we merge their official code with our functions and report the reproduced results. Perplexity of other models and comparison with other baselines can be found in Appendix B.

| LLaMA-7B | 4 bit | | | 3.x bit | | | 3 bit | | | 2.x bit | | |
|---|---|---|---|---|---|---|---|---|---|---|---|---|
| | Bit | PPL | | Bit | PPL | | Bit | PPL | | Bit | PPL | |
| | | Wiki | C4 | | Wiki | C4 | | Wiki | C4 | | Wiki | C4 |
| FP16 | - | 5.68 | 7.08 | - | 5.68 | 7.08 | - | 5.68 | 7.08 | - | 5.68 | 7.08 |
| SqueezeLLM | 4 | **5.79** | 7.21 | 3.24 | 6.13 | 7.56 | 3 | 6.32 | 7.75 | 2.23 | 11.32 | 15.69 |
| OmniQuant | 4 | 5.86 | 7.34 | 3.24 | 6.15 | 7.75 | 3 | 6.48 | 8.19 | 2.25 | 9.72 | 12.79 |
| QuIP# | 4 | 5.83 | 7.25 | - | - | - | 3 | 6.29 | 7.82 | 2 | 9.95 | 11.70 |
| SKIM | 4 | **5.79** | **7.20** | 3.2 | **6.07** | **7.52** | 3 | **6.21** | **7.68** | 2.25 | **8.99** | **11.00** |

| LLaMA2-7B | 4 bit | | | 3.x bit | | | 3 bit | | | 2.x bit | | |
|---|---|---|---|---|---|---|---|---|---|---|---|---|
| | Bit | PPL | | Bit | PPL | | Bit | PPL | | Bit | PPL | |
| | | Wiki | C4 | | Wiki | C4 | | Wiki | C4 | | Wiki | C4 |
| FP16 | - | 5.47 | 6.97 | - | 5.47 | 6.97 | - | 5.47 | 6.97 | - | 5.47 | 6.97 |
| SqueezeLLM | 4 | 5.62 | 7.12 | 3.24 | 5.96 | 7.51 | 3 | 6.18 | 7.72 | 2.23 | - | - |
| OmniQuant | 4 | 5.74 | 7.35 | 3.25 | 6.03 | 7.75 | 3 | 6.58 | 8.65 | 2.25 | 11.06 | 15.02 |
| QuIP# | 4 | 5.66 | 7.17 | - | - | - | 3 | 6.19 | 7.85 | 2 | 12.30 | 14.80 |
| SKIM | 4 | **5.60** | **7.11** | 3.2 | **5.91** | **7.48** | 3 | **6.09** | **7.66** | 2.25 | **10.10** | **12.42** |

## 5.2. Perplexity Results

The results of our SKIM method applied to the LLaMA-7B/13B models are presented in Table 1. Our method consistently outperforms alternatives across various configurations, demonstrating its versatility and effectiveness. Notably, with 3-bit quantization, our method achieves a significant reduction in perplexity, narrowing the performance gap between full precision and 3-bit quantized models by 18.5% on LLaMA-7B and 15.7% on LLaMA-13B. Furthermore, our 3.2-bit model even surpasses others that operates at a slightly higher bit level, further highlighting the effectiveness of our method. These substantial reductions are also evident in other models, as detailed in Appendix B.

## 5.3. Benchmark Result

We evaluate our proposed method on various benchmarks, including PIQA, ARC-Challenge, ARC-Easy, and MMLU.

| Objective | Perplexity | | Objective | Perplexity |
|---|---|---|---|---|
| *L-full* | - | | *L-full* | 6.24 |
| *S-diag* | 6.33 | | *S-diag* | 6.27 |
| *L-diag* | 6.36 | | *L-diag* | 6.29 |

*Table 2.* Comparison of effectiveness between different objectives. On the left, we present the perplexity results for weighted K-means clustering, excluding the results of *L-full* due to its non-conformity to the element-wise sum. On the right, we display the perplexity results for the scaling vector scenario.

Since SqueezeLLM does not provide benchmarking results for LLaMA models, we benchmark the quantized model generated by their official code to ensure consistency and reliability in our evaluation. As highlighted in Table 3, our SKIM method demonstrates a notable enhancement in the performance of the quantized LLaMA-7B across all tested benchmarks and both INT3 and INT4 precision levels. Specifically, in the 3-bit setting, SKIM outperforms SqueezeLLM in all benchmarks, showcasing its effectiveness in preserving knowledge retention during quantization. Additionally, in the 4-bit configuration, SKIM achieves a higher average accuracy level as well, reinforcing the method's robustness across various settings. As a result, this evaluation, covering diverse domains, clearly evidences the consistent performance improvements of our method.

## 5.4. Ablation Study

### 5.4.1. EFFECTIVENESS OF OPTIMIZATION OBJECTIVES

As mentioned earlier, different forms and contents of optimization objectives can lead to varying levels of effectiveness. Here, effectiveness refers to how much the final loss or perplexity is positively influenced by optimizing a specific target. We selected two scenarios to evaluate the objectives: the sampled weights for K-means clustering and the loss calculation for optimizing the scaling vector. In the K-means clustering scenario, both mixed precision and the scaling vector are disabled, while in the scaling vector scenario, we use *S-diag* as weights for clustering and disable

*Table 3.* Comparison of averaged benchmark accuracy on LLaMA2-7B. Tested benchmarks include PIQA, ARC-C, ARC-E and MMLU.

| Method | Bit | PIQA | ARC-C | ARC-E | MMLU | Avg |
|---|---|---|---|---|---|---|
| LLaMA2-7B | 16 | 77.0% | 42.1% | 64.5% | 45.9% | 57.4% |
| SqueezeLLM | 3 | 75.6% | 38.8% | 61.1% | 41.3% | 54.2% |
| SKIM | 3 | 76.1% | 40.6% | 61.3% | 42.3% | 55.1% |
| SqueezeLLM | 4 | 77.0% | 41.8% | 63.8% | 45.1% | 56.9% |
| SKIM | 4 | 76.9% | 42.5% | 64.6% | 45.4% | 57.3% |

mixed precision. We employ 3-bit quantization and final perplexity to illustrate effectiveness. Detailed results can be found in Table 2, aligning well with our theoretical analysis. Evaluation of the *S-full* form is excluded due to its quadratic complexity, which renders it impractical.

### 5.4.2. ITERATIVE OPTIMIZATION

Separating the optimization and assigning different objectives for different processes does not lead to fluctuations in perplexity, as illustrated in Figure 5. Disabling scaling gives us the grouping-only curve, while enabling it produces the grouping-scaling curve. The consistent drop in perplexity after enabling the vector validates the synchronization between the targets. Regarding the number of iterations, we empirically find that the perplexity reduction brought by extra iterations is much more modest compared to the significant decrease observed after the first iteration between grouping and scaling. Additionally, increasing the number of iterations can sometimes lead to overfitting, which causes slight fluctuations in perplexity. Therefore, we are confident to set the maximum number of iterations to one.

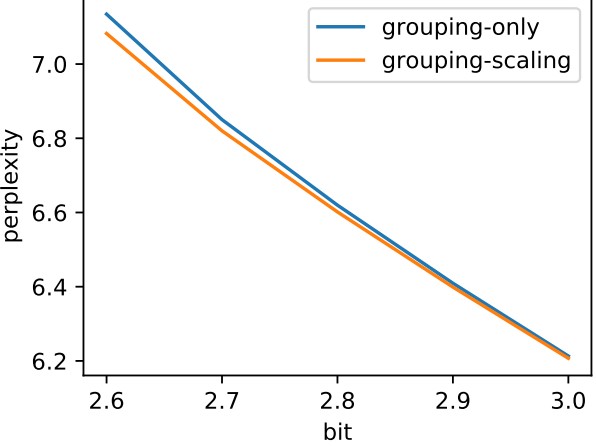

*Figure 5.* Perplexity variation after enabling scaling vector. Perplexity consistently decreases when additional optimization on the scaling vector is applied.

### 5.5. Memory Efficiency

Following the approach outlined in SqueezeLLM (Kim et al., 2023), we evaluated the Peak Memory Usage of our SKIM method when generating 64 tokens, as detailed in Table 4. Our method achieved lower peak memory usage even at the same bit level due to the utilization of lower precision for cluster centroids. Additionally, the 3.2-bit quantized model offers comparable memory savings to the 3.24-bit model from SqueezeLLM, while also outperforming it. Beyond memory efficiency, our approach stands out for its ability to break the fixed bit grid. Given a specific GPU capacity constraint, users can select the maximum bit level to fully exploit the machine. For more detailed information, please refer to Appendix C.

| LLaMA-7B | 3bit | 3.xbit |
|---|---|---|
| FP16 | 12.72GB | |
| Squeeze | 3.01GB | 3.26GB |
| SKIM | 2.98GB | 3.13GB |

| LLaMA-13B | 3bit | 3.xbit |
|---|---|---|
| FP16 | 24.63GB | |
| Squeeze | 5.45GB | 5.88GB |
| SKIM | 5.41GB | 5.69GB |

*Table 4.* Memory efficiency with LLaMA-7B and LLaMA-13B.

## 6. Conclusion

We propose **S**caled **K**-means clustering w**I**th **M**ixed Precision (SKIM), an effective posting-training and weight-only quantization method. Building on previous non-uniform quantization methods, SKIM further incorporate two novel techniques: Adaptive Mixed Precision and Trainable Scaling Vector. Our method is evaluated across a wide range of models, tasks, and bit levels, consistently outperforming previous state-of-the-art methods. Its memory efficiency and ability to break the fixed bit grid facilitate the deployment of large language models.

## Impact Statement

SKIM advances efficient LLM deployment through adaptive quantization, enabling resource-constrained applications like on-device AI and reducing cloud inference costs. This could democratize access to powerful language models while lowering resources consumption. However, wider accessibility may also lower barriers for malicious actors deploying LLMs at scale. Though SKIM maintains high accuracy, subtle quantization artifacts could propagate biases or errors from base models in safety-critical domains like healthcare or finance. To mitigate risks, we encourage: (1) robustness testing for quantized models in sensitive applications, (2) developing auditing standards for compressed models, and (3) continued research into high and safe machine learning efficiency. While SKIM itself introduces no novel harms, its compression features and efficiency gains warrant responsible deployment protocols. Overall, the benefits of SKIM in improving model accessibility outweigh its risks, but careful consideration of these factors is essential for responsible deployment.

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

# A. Full Algorithm

The full algorithm for our SKIM method is illustrated in Algorithm 3. Note that we provide the simplified version with number of iterations equal to one. Our method includes three main steps: 1. compute the bit allocation (line 2-11); 2. apply channel-wise K-Means Clustering (line 12-15); 3. train the scaling vector (line 16-22). Additionally, error pre-recording (line 2-10) only needs to be executed once for each model, and all KMeans functions are accelerated using multi-processing and shared memory.

---

**Algorithm 3** Overall Algorithm for SKIM

---

**input** Weight $W$, Gradient Square $G$, Bit $bit$, Minimum Available Bit $b_{\min}$, Maximum Available Bit $b_{\max}$
**output** Labels $L$, Centroids $c$, Scaling Vector $\alpha$

1: $n, m = W.shape$
2: $E = [\,0\,]_{n \times (b_{\max} - b_{\min} + 1)} \in R^{n \times (b_{\max} - b_{\min} + 1)}$         *// Error Matrix*
3: **for** $i = 1$ **to** $n$ **do**
4:     **for** $\hat{b} = b_{\min}$ **to** $b_{\max}$ **do**
5:        $result \leftarrow Kmeans(W_{i,:}, weights = G_{i,:}, n\_centriods = 2^{\hat{b}})$    *// apply Kmeans(·) with actual configuration*
6:        $l, c \leftarrow result.labels, result.centroids$
7:        $w^q = replace(c, l)$                           *// replace labels with corresponding centroids*
8:        $L_{i,\hat{b}} = Err(W_{i,:}, w^q)$                    *// pre-record quantization error with Equation 5*
9:     **end for**
10: **end for**
11: $b = alloc\_bit(E, b_{\min}, b_{\max}, n, bit) \in \mathbb{R}^n$          *// compute bit allocation with Algorithm 1*
12: $L = \mathbf{0}_{n \times m} \in \mathcal{R}^{n \times m}$
13: **for** $i = 1$ **to** $n$ **do**
14:     $L_{i,:} \leftarrow Kmeans(W_{i,:}, weights = G_{i,:}, n\_centriods = 2^{b_i}).labels$    *// optimize on grouping towards Equation 4*
15: **end for**
16: Initialize $\boldsymbol{\alpha} = \mathbf{1}_{1 \times m} \in \mathcal{R}^{1 \times m}$
17: **repeat**
18:     $c = calc\_centroids(W * \boldsymbol{\alpha}^{-1}, L)$
19:     $W^q = replace(c, L) * \boldsymbol{\alpha}$        *// replace labels with centroids and unscale to reconstruct the weight*
20:     $loss = Err(W, W^q)$                      *// calculate quantization error with Equation 5*
21:     $loss.backward()$                           *// optimize on scaling*
22: **until** Converged
23: **return** $L, c, \alpha$

---

# B. Additional Experiment Result

## B.1. Perplexity Evaluation on Other Models

Table 5 presents results for other LLaMA models, including LLaMA-30B and LLaMA2-7B. Our method continues to outperform other with larger model size and different model series. Additionally, table 6 contains all results on OPT models, where our method also demonstrates superior performance, highlighting its generalizability. However, on OPT models, mixed precision would greatly reduces the quantization error yet slightly increase the final perplexity. For example, on the *up_proj* layer mixed precision reduces the quantization error by more than 50% but increases the perplexity. We attribute this phenomenon to overfitting on the calibration data, and address it by disabling mixed precision under integer bit levels while initializing the bit allocation with flooring to the specified bits under non-integer levels. Consequently, even with the slight increase in perplexity, our method still provides better or comparable results.

## B.2. Comparation with other baselines

DecoupleQ (Guo et al., 2024) decouples model parameters into integer and floating point parts, achieving state-of-the-art performance in certain low-bit configurations. Table 8 compares our SKIM method with DecoupleQ, and C4 results in excluded as DecoupleQ does not provide corresponding results. SKIM consistently outperforms DecoupleQ across most settings by a significant margin, highlighting its effectiveness. When operating at INT2 precision with additional memory,

*Table 5.* **Quantization Result of other LLaMA models**

| LLaMA-13B | 4 bit | | | 3.x bit | | | 3 bit | | | 2.x bit | | |
|---|---|---|---|---|---|---|---|---|---|---|---|---|
| | Bit | PPL | | Bit | PPL | | Bit | PPL | | Bit | PPL | |
| | | Wiki | C4 | | Wiki | C4 | | Wiki | C4 | | Wiki | C4 |
| FP16 | - | 5.09 | 6.61 | - | 5.09 | 6.61 | - | 5.09 | 6.61 | - | 5.09 | 6.61 |
| SqueezeLLM | 4 | 5.18 | 6.71 | 3.25 | 5.45 | 6.92 | 3 | 5.60 | 7.08 | 2.23 | 8.74 | 12.57 |
| OmniQuant | 4 | 5.21 | 6.76 | 3.25 | 5.44 | 7.05 | 3 | 5.68 | 7.32 | 2.24 | 7.93 | 10.76 |
| QuIP# | 4 | 5.20 | **6.70** | - | - | - | 3 | 5.53 | **6.98** | - | - | - |
| SKIM | 4 | **5.17** | **6.70** | 3.2 | **5.42** | **6.92** | 3 | 5.52 | 7.04 | 2.25 | **7.40** | **9.22** |

| LLaMA-30B | 4 bit | | | 3.x bit | | | 3 bit | | | 2.x bit | | |
|---|---|---|---|---|---|---|---|---|---|---|---|---|
| | Bit | PPL | | Bit | PPL | | Bit | PPL | | Bit | PPL | |
| | | Wiki | C4 | | Wiki | C4 | | Wiki | C4 | | Wiki | C4 |
| FP16 | - | 4.10 | 5.98 | - | 4.10 | 5.98 | - | 4.10 | 5.98 | - | 4.10 | 5.98 |
| SqueezeLLM | 4 | 4.22 | 6.06 | 3.25 | **4.44** | 6.23 | 3 | 4.66 | 6.37 | 2.22 | - | - |
| OmniQuant | 4 | 4.25 | 6.11 | 3.25 | 4.56 | 6.37 | 3 | 4.74 | 6.57 | 2.24 | 6.59 | 9.36 |
| QuIP# | 4 | 4.23 | 6.06 | - | - | - | 3 | 4.54 | 6.29 | 2 | **5.80** | 7.55 |
| SKIM | 4 | **4.20** | **6.05** | 3.2 | 4.46 | **6.22** | 3 | 4.53 | 6.31 | 2.25 | **5.80** | **7.49** |

*Table 6.* **Quantization Result of OPT models**

| OPT-2.7B | 4 bit | | | 3.x bit | | | 3 bit | | | 2.x bit | | |
|---|---|---|---|---|---|---|---|---|---|---|---|---|
| | Bit | PPL | | Bit | PPL | | Bit | PPL | | Bit | PPL | |
| | | Wiki | C4 | | Wiki | C4 | | Wiki | C4 | | Wiki | C4 |
| FP16 | - | 12.47 | 13.17 | - | 12.47 | 13.17 | - | 12.47 | 13.17 | - | 12.47 | 13.17 |
| SqueezeLLM | 4.07 | 12.80 | 13.38 | 3.25 | 13.43 | **13.88** | 3 | 13.85 | 14.45 | - | - | - |
| OmniQuant | 4 | 12.76 | 13.58 | 3.24 | 13.18 | 14.15 | 3 | 13.80 | 14.93 | 2.25 | **18.13** | 21.11 |
| SKIM | 4 | **12.72** | **13.35** | 3.2 | **13.34** | 13.92 | 3 | **13.66** | **14.21** | 2.25 | 19.79 | **19.96** |

| OPT-6.7B | 4 bit | | | 3.x bit | | | 3 bit | | | 2.x bit | | |
|---|---|---|---|---|---|---|---|---|---|---|---|---|
| | Bit | PPL | | Bit | PPL | | Bit | PPL | | Bit | PPL | |
| | | Wiki | C4 | | Wiki | C4 | | Wiki | C4 | | Wiki | C4 |
| FP16 | - | 10.12 | 11.20 | - | 10.12 | 11.20 | - | 10.12 | 11.20 | - | 10.12 | 11.20 |
| SqueezeLLM | 4 | 11.03 | 11.85 | 3.26 | 11.31 | **12.18** | 3 | 11.70 | 12.44 | - | - | - |
| OmniQuant | 4 | 11.03 | 11.97 | 3.25 | **11.27** | 12.31 | 3 | 11.65 | 12.78 | 2.25 | **14.43** | 16.67 |
| SKIM | 4 | **11.02** | **11.84** | 3.2 | **11.27** | 12.20 | 3 | **11.46** | **12.39** | 2.25 | 14.79 | **16.01** |

SKIM provides comparable performance to DecoupleQ at the same bit level. However, with just a slight increase in memory-specifically, 0.05 bits-our method surpasses DecoupleQ by a considerable margin. We present results with this minimal extra memory to illustrate the flexibility of our approach.

ABQ-LLM (Zeng et al., 2024) proposes a block-wise distribution correlation and compensation schema for Post-Training Quantization, demonstrating strong performance at lower bit-widths as well. We report perplexity results on LLaMA-7B and LLaMA-13B models for INT4 and INT3 settings, as these overlap with our experiment data. As shown in Table 8, SKIM outperforms ABQ-LLM in both the INT4 and INT3 configurations.

*Table 7.* **Comparation with DecoupleQ. Perplexity on WikiText2 is reported in this table.**

| Method | 4 bit | | 3 bit | | 2.x bit | |
|---|---|---|---|---|---|---|
| | Bits | PPL($\downarrow$) | Bits | PPL($\downarrow$) | Bits | PPL($\downarrow$) |
| LLaMA-7B | - | 5.68 | - | 5.68 | - | 5.68 |
| DecoupleQ | 4 | 5.85 | 3 | 6.38 | 2.25 | 8.65 |
| SKIM | 4 | **5.79** | 3 | **6.70** | 2.30 | **8.64** |
| LLaMA-13B | - | 5.09 | - | 5.09 | - | 5.09 |
| DecoupleQ | 4 | 5.21 | 3 | 5.60 | 2.25 | 7.25 |
| SKIM | 4 | **5.17** | 3 | **5.52** | 2.30 | **7.11** |
| LLaMA-30B | - | 4.10 | - | 4.10 | - | 4.10 |
| DecoupleQ | 4 | 4.24 | 3 | 4.67 | 2.25 | 6.04 |
| SKIM | 4 | **4.20** | 3 | **4.57** | 2.30 | **5.66** |

*Table 8.* **Comparison with ABQ-LLM. Perplexity on both WikiText2 and C4 dataset is reported.**

| LLaMA-7B | 4 bit | | 3 bit | |
|---|---|---|---|---|
| | Wiki | C4 | Wiki | C4 |
| FP16 | 5.68 | 7.08 | 5.68 | 7.08 |
| ABQ-LLM | 5.83 | 7.29 | 6.29 | 8.01 |
| SKIM | **5.79** | **7.20** | **6.21** | **7.68** |

| LLaMA-13B | 4 bit | | 3 bit | |
|---|---|---|---|---|
| | Wiki | C4 | Wiki | C4 |
| FP16 | 5.09 | 6.61 | 5.09 | 6.61 |
| ABQ-LLM | 5.19 | 6.75 | 5.56 | 7.24 |
| SKIM | **5.17** | **6.70** | **5.52** | **7.04** |

# C. Memory Usage With Varying Bit Levels

Figure 6 illustrates the correlation between memory usage and bit levels. The peak memory usage increases linearly with the bit level, which facilitates the selection of an appropriate level to meet specific memory capacity requirements and to maximize the model's performance under a specific machine.

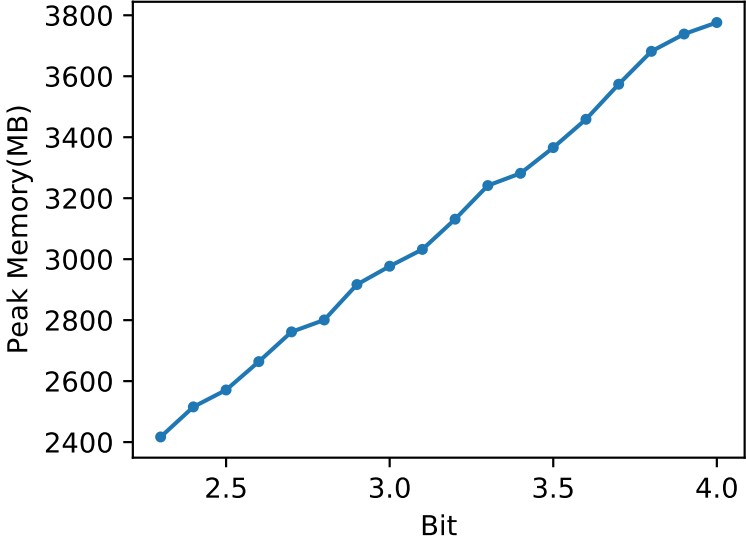

*Figure 6.* The actual peak memory usage of LLaMA-7B when generating 64 tokens.

# D. Quantization Cost

### D.1. Memory Demands

As a post-training quantization method, SKIM inherits the positive characteristics of memory savings. Although it involves training, this process is layer-wise, and the scaling vector is the only trainable parameter. As a result, the memory requirements during the quantization phase are significantly lower than those during inference. For instance, quantizing LLaMA-7B requires peak memory usage of less than 8GB, which is well within the memory capacity of most modern GPUs.

### D.2. Quantization Time

In terms of K-means clustering, we leverage parallel execution and shared memory to enhance efficiency. Speed for both error matrix pre-recording and K-means quantization benefits from these improvements. To record the errors of LLaMA-7B using our framework, the process takes less than half an hour on dual AMD EPYC processors, which feature a total of 128 cores and 256 threads. And once the errors are recorded, they can be utilized for quantization at any specified bit level without needing to repeat the process. For K-means quantization, we can process approximately 8000 rows per second under the same machine conditions, while a transformer block in LLaMA-7B only contains 42496 rows.

Regarding the packing and unpacking phases, by avoiding sparse matrices and consolidating all rows with same bit level, the packing phase is significantly faster than that of SqueezeLLM. It takes about one minute to pack our quantized LLaMA-7B, whereas SqueezeLLM typically exceeds five minutes.

Overall, the entire process for quantizing LLaMA-7B takes around one hour with dual AMD EPYC processors and an RTX 3090 GPU. Compared to OmniQuant, our method is more efficient as we break the block-wise training to layer-wise one. And compared to SqueezeLLM, although the time consumption is slightly higher due to the computations involved in bit allocation and scaling vector training, it still remains comparable, ensuring high efficiency.

