# OpenReview forum: "SKIM: Any-bit Quantization Pushing The Limits of Post-Training Quantization"
_ICML.cc/2025/Conference — ICML 2025 poster_

### Official Review · Reviewer_ABnm · 2025-03-08

**Overall Recommendation:** 3

**Summary:**

This paper introduces a mixed-precision weight-only quantization method for large language models (LLMs). The authors propose a greedy algorithm to allocate bitwidths across weight channels, followed by K-means clustering based on the assigned bitwidths. To enhance performance, the authors further incorporate a trainable scaling vector to refine the non-differentiable clustering process. Experiments on LLaMA models demonstrate the effectiveness of the proposed approach.

**Claims And Evidence:**

Yes.

**Essential References Not Discussed:**

Please see my detailed comments in the following.

**Experimental Designs Or Analyses:**

Yes.

**Methods And Evaluation Criteria:**

Yes.

**Other Comments Or Suggestions:**

Please see my detailed comments.

**Other Strengths And Weaknesses:**

Strengthens:

The proposed method is simple yet intuitive. The authors introduce a preliminary unified perspective on layer-wise and sensitivity-based quantization, offering valuable insights that enhance the reader’s understanding.

Weaknesses:

The paper lacks a thorough discussion and experimental comparisons with several existing weight quantization methods, which would help situate the proposed approach within the broader context of related work. Please see my detailed comments.

**Questions For Authors:**

1.	In Section 4.1, the authors state that the effectiveness of different objectives ranks from best to worst as S-full, L-full, S-diag, and L-diag. Could the authors clarify whether S-diag performs better than L-full? Additionally, it would be helpful to explain how this ranking is determined.

2.	In Section 4.2, the authors propose assigning different bitwidths to different weight channels, which effectively reduces quantization errors. However, during inference, the quantized weights must be dequantized to FP16 for computation. Could the authors clarify whether dequantization in mixed-precision settings is slower than in uniform-precision settings? If so, to what extent?

3.	The proposed method uses mixed-precision quantization to improve the accuracy of the quantized models. Several works [1][2] have also explored mixed-precision quantization. It would be beneficial for the authors to include a more detailed discussion of these related approaches, highlighting the differences.

4.	The experimental comparisons are insufficient, as several state-of-the-art weight-only quantization methods are not included, such as [3][4][5], among others.

5.	The settings of the compared quantization methods are not clearly specified. For example, do the results reported for OmniQuant correspond to group-wise or channel-wise quantization?

6.	The experimental results on LLaMA, LLaMA2, and OPT are outdated. It would be beneficial for the authors to include additional results on LLaMA3 [6].

7.	The experimental results based solely on MMLU are insufficient. The authors should include additional commonly used datasets, such as LAMBADA [7], ARCEasy (ArcE) [8], and PIQA [9], which are frequently used in the literature to evaluate the performance of quantized models.

Reference:

[1] SliM-LLM: Salience-Driven Mixed-Precision Quantization for Large Language Models. arXiv 2024.

[2] CLAQ: Pushing the Limits of Low-Bit Post-Training Quantization for LLMs. arXiv 2024.

[3] AWQ: Activation-aware Weight Quantization for LLM Compression and Acceleration. MLSys 2024.

[4] QuIP: 2-Bit Quantization of Large Language Models With Guarantees. NeurIPS 2023.

[5] QuIP#: Even Better LLM Quantization with Hadamard Incoherence and Lattice Codebooks. ICML 2024.

[6] The Llama 3 Herd of Models. arXiv 2024.

[7] The LAMBADA dataset: Word prediction requiring a broad discourse context. ACL 2016.

[8] A systematic classification of knowledge, reasoning, and context within the ARC dataset. ACL 2018.

[9] Piqa: An algebra for querying protein data sets. International Conference on Scientific and Statistical Database Management 2003.

**Relation To Broader Scientific Literature:**

The paper proposes a mixed-precision quantization method for LLMs. However, it lacks a thorough discussion and experimental comparisons with existing methods, which would help contextualize the contributions within the broader literature.

**Theoretical Claims:**

No theoretical claims.

---

> ### Author Rebuttal · Authors · 2025-03-31
>
> 1. Clarification on Objective Rankings:
> The ranking of objectives (S-full > L-full > S-diag > L-diag) is elaborated in Section 4.1, where we demonstrate that L-full outperforms S-diag. This is because S-diag simplifies computation via a diagonal assumption, introducing a certain degree of bias compared to the full form. Practically, Section 5.4.1 demonstrates that L-full outperforms S-diag in perplexity, validating our analysis.
>
> 2. Dequantization Efficiency in Mixed-Precision Settings:
> In mixed-precision implementations, we group channels with identical bitwidths and track their original indices to leverage GPU parallelism. Although dequantization introduces minor overhead compared to uniform-precision methods, this overhead is smaller than the computational cost of sparse tensor operations (e.g., in SqueezeLLM) and is justified by significant accuracy improvements. Our method performs similarly to SqueezeLLM latency-wise and faster than it throughput-wise.
>
> 3. Differentiation from Prior Mixed-Precision Work:
> Existing methods like SqueezeLLM, SliM-LLM, and CLAQ mix only a limited set of predefined precisions. For instance, SqueezeLLM combines INT3/4 with FP16, while SliM-LLM and CLAQ mix INT X with INT X+k and also include INT X-k for compensation. In contrast, our framework adaptively assigns arbitrary bit widths (e.g., mixing 1~4bit) across channels without relying on fixed thresholds or compensation rules. This flexibility broadens the scope of discussion and enables finer-grained error minimization.
>
> 4. Expanded Baseline Comparisons:
> We have added comparisons with QuIP#, a state-of-the-art baseline, which outperforms AWQ and QuIP. We choose the pure-PTQ version of QuIP# (QuIP# with fine-tuning consumes greater computational resources; for example, it requires 50 GPU hours to quantize LLaMA-70B on an 8 GPU node) and the the result is included in the following table:
>
> | LLaMA2-7B      | 4-bit |          |          | 3.x-bit |          |          | 3-bit |          |          | 2.x-bit |           |           |
> | ---------- | :----: | -------- | -------- | :----: | -------- | -------- | :----: | -------- | -------- | :----: | --------- | --------- |
> |            | Bit   | Wiki     | C4       | Bit     | Wiki     | C4       | Bit   | Wiki     | C4       | Bit     | Wiki      | C4        |
> | FP16       | -     | 5.47     | 6.97     | -       | 5.47     | 6.97     | -     | 5.47     | 6.97     | -       | 5.47      | 6.97      |
> | SqueezeLLM | 4     | 5.62     | 7.12     | 3.24    | 5.96     | 7.51     | 3     | 6.18     | 7.72     | 2.23    | -         | -         |
> | OmniQuant  | 4     | 5.74     | 7.35     | 3.25    | 6.03     | 7.75     | 3     | 6.58     | 8.65     | 2.25    | 11.06     | 15.02     |
> | QuIP#      | 4     | 5.66     | 7.17     | -       | -        | -        | 3     | 6.19     | 7.85     | 2       | 12.30     | 14.80     |
> | **SKIM**   | 4     | **5.60** | **7.11** | 3.2     | **5.91** | **7.48** | 3     | **6.09** | **7.66** | 2.25    | **10.10** | **12.42** |
>
> Note that QuIP# officially supports only integer-bit quantization (e.g., INT2); thus, we compare its INT2 results with our 2.x-bit performance.
>
> 5. OmniQuant Implementation Details:
> Under non-integer bitwidths(eg: 3.x-bits), OmniQuant is evaluated under its group-wise quantization setting (eg: 128 elements per group), as specified in their official implementation. For integer bits (e.g., 3/4-bit), per-channel quantization is used, same as our method. Previous works commonly adopt this setting.
>
> 6. LLaMA3 Benchmarking:
> We focus on LLaMA-1/2 and OPT models to align with established baselines (SqueezeLLM, OmniQuant, and baselines you mentioned), as LLaMA3 results are not yet widely reported in quantization literature. This ensures apples-to-apples comparisons. We will be happy to include LLaMA3 in future work once baselines adopt it.
>
> 7. Expanded Evaluation Metrics:
> Following suggestions, we have added results on ARC-E, ARC-C, and PIQA. Below is the detailed result and our method maintains its consistent superiority.
> | Method                | Precision |   PIQA    |   ARC-C   |   ARC-E   |   MMLU    |    Avg    |
> | --------------------- | :-------: | :-------: | :-------: | :-------: | :-------: | :-------: |
> | **LLaMA2-7B**         |   FP16    |   77.0%   |   42.1%   |   64.5%   |   45.9%   |   57.4%   |
> | **INT3 Quantization** |           |           |           |           |           |           |
> | SKIM                  |   INT3    | **76.1%** | **40.6%** | **61.3%** | **42.3%** | **55.1%** |
> | SqueezeLLM            |   INT3    |   75.6%   |   38.8%   |   61.1%   |   41.3%   |   54.2%   |
> | **INT4 Quantization** |           |           |           |           |           |           |
> | SKIM                  |   INT4    |   76.9%   | **42.5%** | **64.6%** | **45.4%** | **57.3%** |
> | SqueezeLLM            |   INT4    | **77.0%** |   41.8%   |   63.8%   |   45.1%   |   56.9%   |

---

> > ### Comment · Reviewer_ABnm · 2025-04-07
> >
> > The authors have adequately addressed my concerns, so I have decided to raise my score.

---

### Official Review · Reviewer_eA4i · 2025-03-14

**Overall Recommendation:** 3

**Summary:**

The paper introduces SKIM (Scaled K-means clustering wIth Mixed precision), a novel post-training quantization method for Large Language Models (LLMs) that supports any-bit quantization. The key contributions include a greedy algorithm for optimal bit allocation across weight channels, addressing the significant disparity in data distribution across channels, and a trainable scaling vector for non-differentiable K-means clustering, which regularizes column-wise differences and complements the mixed-precision method.

**Claims And Evidence:**

The mixed-precision technique with greedy bit allocation improves quantization performance by adaptively allocating bits per channel. This is demonstrated through experiments showing reduced quantization errors and improved perplexity across different bit levels. The trainable scaling vector effectively regularizes data across columns and improves quantization accuracy. Experimental results confirm that incorporating the scaling vector leads to consistent drops in perplexity.

**Essential References Not Discussed:**

The paper could benefit from discussing more recent works on adaptive quantization and hardware-aware optimizations for LLM inference.

**Experimental Designs Or Analyses:**

The authors evaluate on multiple LLMs (LLaMA, LLaMA2, OPT) to demonstrate generalizability, and they compare against relevant baselines (SqueezeLLM, OmniQuant) under similar conditions.

**Methods And Evaluation Criteria:**

The proposed methods are appropriate for the problem of post-training quantization of LLMs. The greedy algorithm for bit allocation addresses the observed disparity in quantization errors across channels, making sense for resource optimization. And the trainable scaling vector provides a solution for regularizing non-differentiable K-means clustering, which is a common challenge in quantization methods.

**Other Comments Or Suggestions:**

N/A

**Other Strengths And Weaknesses:**

The article is clearly structured and written. The methods are well presented. If possible, more illustrations or ablation studies would be helpful.

**Questions For Authors:**

N/A

**Relation To Broader Scientific Literature:**

Builds on previous work in non-uniform quantization methods, particularly K-means clustering approaches. Extends the research on post-training quantization for LLMs, addressing limitations of existing methods that often experience significant performance drops at lower precision levels. Contributes to the growing body of work on efficient inference methods for LLMs, focusing on memory and computational efficiency.

**Theoretical Claims:**

The authors provide a detailed analysis of layer-wise and sensitivity-based quantization objectives, showing how they can be transformed into unified structures. The derivation of the mixed-precision problem as a bit-constrained sum minimization is logically sound, though a more in-depth examination of the dynamic programming algorithm's limitations would strengthen this section.

---

> ### Author Rebuttal · Authors · 2025-03-31
>
> 1. More Information about the Greedy Algorithm:
> Through empirical validation, we have demonstrated the effectiveness and efficiency of the greedy algorithm:
>
>  - Time Efficiency: The cpp/jit implementation of the dynamic programming algorithm remains slow. For a single block in LLaMA-7B, it requires several minutes of computation. Given that LLaMA-7B has 32 layers with 7 blocks per layer, this algorithm struggles to scale to large models. In contrast, the greedy algorithm, even when implemented in Python, only requires approximately 3 seconds per block, ensuring computational efficiency.
>
>  - Practical Effectiveness: To guarantee a near-optimal solution, we employ two different initialization points for the greedy algorithm and select the superior solution. Empirical validation on the first layer of LLaMA-7B shows that the average error between the suboptimal solution from the greedy algorithm and the true optimal solution is below 2%, confirming its capability to deliver sufficiently high-quality results.
>
>
> 2. Additional experiments:
> We further validated our method by comparing it with more recent works (e.g., adding QuIP# as a new baseline) and extended benchmarking on tasks like PiQA and ARC. These results further validates the effectiveness of our method and detailed results are shown below:
>
> | Method                | Precision |   PIQA    |   ARC-C   |   ARC-E   |   MMLU    |    Avg    |
> | --------------------- | :-------: | :-------: | :-------: | :-------: | :-------: | :-------: |
> | **LLaMA2-7B**         |   FP16    |   77.0%   |   42.1%   |   64.5%   |   45.9%   |   57.4%   |
> | **INT3 Quantization** |           |           |           |           |           |           |
> | SKIM                  |   INT3    | **76.1%** | **40.6%** | **61.3%** | **42.3%** | **55.1%** |
> | SqueezeLLM            |   INT3    |   75.6%   |   38.8%   |   61.1%   |   41.3%   |   54.2%   |
> | **INT4 Quantization** |           |           |           |           |           |           |
> | SKIM                  |   INT4    |   76.9%   | **42.5%** | **64.6%** | **45.4%** | **57.3%** |
> | SqueezeLLM            |   INT4    | **77.0%** |   41.8%   |   63.8%   |   45.1%   |   56.9%   |
>
> | LLaMA2-7B      | 4-bit |          |          | 3.x-bit |          |          | 3-bit |          |          | 2.x-bit |           |           |
> | ---------- | :-----: | -------- | -------- | :-------: | -------- | -------- | :----: | -------- | -------- | :-------: | --------- | --------- |
> |            | Bit   | Wiki     | C4       | Bit     | Wiki     | C4       | Bit   | Wiki     | C4       | Bit     | Wiki      | C4        |
> | FP16       | -     | 5.47     | 6.97     | -       | 5.47     | 6.97     | -     | 5.47     | 6.97     | -       | 5.47      | 6.97      |
> | SqueezeLLM | 4     | 5.62     | 7.12     | 3.24    | 5.96     | 7.51     | 3     | 6.18     | 7.72     | -    | -         | -         |
> | OmniQuant  | 4     | 5.74     | 7.35     | 3.25    | 6.03     | 7.75     | 3     | 6.58     | 8.65     | 2.25    | 11.06     | 15.02     |
> | QuIP#      | 4     | 5.66     | 7.17     | -       | -        | -        | 3     | 6.19     | 7.85     | 2       | 12.30     | 14.80     |
> | **SKIM**   | 4     | **5.60** | **7.11** | 3.2     | **5.91** | **7.48** | 3     | **6.09** | **7.66** | 2.25    | **10.10** | **12.42** |

---

### Official Review · Reviewer_CLBX · 2025-03-16

**Overall Recommendation:** 3

**Summary:**

The work describes a post-training quantization technique that allows for non-integer size quantization of model parameters.

## update after rebuttal

The rebuttal process helped to address my concerns and I thank the authors for their supporting answer(s). As a result I increased my score to 3, in hope the disscussed additions will be added to the final submission.

**Claims And Evidence:**

The claim that this beats baselines is wrong as it only beats the presented baselines. A recently presented technique (NeurIPS2024) named ShiftAddLLM has shown to give better perplexity / quantization ratios for 2 and 3 bits (for reference: Table 3 lists results using LLaMa models). The work of ShiftAddLLM may not official list as PTQ technique, although it clear is. But I view missing reference and exclusion of a better working approach published earlier to be a major defect of a submission.

**Essential References Not Discussed:**

ShiftAddLLM: https://arxiv.org/abs/2406.05981 with code at https://github.com/GATECH-EIC/ShiftAddLLM

**Experimental Designs Or Analyses:**

The improvements to presented baselines for >3 bits are minor.

**Methods And Evaluation Criteria:**

An important, better baseline was omitted from the comparison.

**Other Comments Or Suggestions:**

None

**Other Strengths And Weaknesses:**

None

**Questions For Authors:**

None

**Relation To Broader Scientific Literature:**

Non-integer size quantization of large sets of integers/floats may help other research areas apart from ML as well. I didn't check, but the idea may have been presented elsewhere already.

**Theoretical Claims:**

The theoretical claims were all sound to me.

---

> ### Author Rebuttal · Authors · 2025-03-31
>
> 1. Comparison with ShiftAddLLM:
> Thank you for the suggestion. We have now included a comparison with ShiftAddLLM. Overall, our method is competitive compared to ShiftAddLLM. In fact, ShiftAddLLM uses a smaller group size (128) rather than channel-wise quantization when quantizing LLaMA, so its precision should be classified as non-integer level. ShiftAddLLM does not provide detailed explanations about the precision of the scaling factors or additional overhead, so we compare its performance under the common practice of group size 128. We represented the perplexity of quantized LLaMA-7B on Wikitext2 in the table below. our method performs better at 3-bit, while being comparable at 2-bit. Moreover, by slightly increasing the bit level using the any-bit feature, significant improvements can be achieved. Additionally, beyond perplexity (PPL), our method achieves state-of-the-art performance on some widely-adopted benchmarks (see the table under answer 2), with some benchmarks even approaching FP16 performance. Those empirical results demonstrate that our method achieves comparable or superior performance to ShiftAddLLM. Furthermore, we want to emphasize the superior quantization efficiency of our method. While ShiftAddLLM requires approximately 20 GPU-hours to quantize the LLaMA2-7B, our approach completes the same process in under one hour. This dramatic difference in computational overhead becomes even more significant when scaling to larger models, making our method more practical for real-world applications.
> | Method         | Bit | Group Size | PPL (Wikitext2) |
> |----------------|-----------|------------|-----------------|
> | FP16 (LLaMA)  | 16        | -          | 5.68            |
> | ShiftAddLLM    | 3.0       | 128        | 6.04            |
> | SKIM (Ours)   | 3.25      | Channel-wise| **6.02**        |
> | ShiftAddLLM    | 2.0       | 128        | 7.98            |
> | SKIM (Ours)    | 2.25      | Channel-wise| 8.44            |
> | SKIM (Ours)    | 2.4       | Channel-wise| **7.76**        |
>
>
>
> 2. Improvements for >3 Bits:
> In fact, the improvements shown in perplexity are nonlinear—smaller improvements at lower PPL can still lead to notable performance gains. For example, when quantizing LLaMA2-7B, our compressed model demonstrates significant improvements on PiQA, ARC, and MMLU, as detailed in the table below. Meanwhile, the smaller PPL improvement at 3.x-bit is because, for consistency, we compressed the model to 3.2 bits rather than higher bit levels like 3.25 or above used by some other methods. For 4-bit, we set the maximum available bit to 4 to ensure quantization efficiency, effectively disabling mixed precision. If we were to use higher bit levels for 3.x-bit (e.g., 3.25 or 3.3 bits), PPL could be reduced by an additional 50%-100% (from 6.13 to 6.07 to 6.02), making the improvement more pronounced. Similarly, for 4-bit, if we relaxed the restriction, the gap to FP16 could be reduced from 0.13 to 0.11. This 0.02 reduce is relatively considerable for the INT4 level which is quite close to the original FP16 result.
> | Method                | Precision |   PIQA    |   ARC-C   |   ARC-E   |   MMLU    |    Avg    |
> | --------------------- | :-------: | :-------: | :-------: | :-------: | :-------: | :-------: |
> | **LLaMA2-7B**         |   FP16    |   77.0%   |   42.1%   |   64.5%   |   45.9%   |   57.4%   |
> | **INT3 Quantization** |           |           |           |           |           |           |
> | SKIM                  |   INT3    | **76.1%** | **40.6%** | **61.3%** | **42.3%** | **55.1%** |
> | SqueezeLLM            |   INT3    |   75.6%   |   38.8%   |   61.1%   |   41.3%   |   54.2%   |
> | **INT4 Quantization** |           |           |           |           |           |           |
> | SKIM                  |   INT4    |   76.9%   | **42.5%** | **64.6%** | **45.4%** | **57.3%** |
> | SqueezeLLM            |   INT4    | **77.0%** |   41.8%   |   63.8%   |   45.1%   |   56.9%   |
>
>
>
> 3. Clarification on Any-Bit:
> Previous methods do include some works that support any-bit due to mixed precision, but most fail to demonstrate "continuity" in bit usage. For example, SqueezeLLM allows increasing the precision of certain elements, but if an extra available bit is added to INT3, its performance would be worse than INT4 rather than matching it. In contrast, our any-bit method, based on allocation, can fully utilize the given bit budget, making it unique and practical.

---

### Decision · Program_Chairs · 2025-05-01

**Decision:**

Accept (poster)

**Comment:**

There is a strong consensus among the reviewers that this paper should be accepted, so I'm going along with that. But I think it's missing comparisons to some more recent related work that should be included: if you're going to compare to QuIP# you should also (or instead) compare to QTIP ("QTIP: Quantization with Trellises and Incoherence Processing," NeurIPS 2024), which is a more recent and better method from the same family (a more SOTA baseline than QuIP#). SpinQuant ("SpinQuant: LLM quantization with learned rotations") also seems relevant.